# Chromatic micromaps in primary visual cortex

Soumya Chatterjee[1,2], Kenichi Ohki[1,3] & R. Clay Reid [1,2✉]

The clustering of neurons with similar response properties is a conspicuous feature of neocortex. In primary visual cortex (V1), maps of several properties like orientation preference are well described, but the functional architecture of color, central to visual perception in trichromatic primates, is not. Here we used two-photon calcium imaging in macaques to examine the fine structure of chromatic representation and found that neurons responsive to spatially uniform, chromatic stimuli form unambiguous clusters that coincide with blobs. Further, these responsive groups have marked substructure, segregating into smaller ensembles or micromaps with distinct chromatic signatures that appear columnar in upper layer 2/3. Spatially structured chromatic stimuli revealed maps built on the same micromap framework but with larger subdomains that go well beyond blobs. We conclude that V1 has an architecture for color representation that switches between blobs and a combined blob/interblob system based on the spatial content of the visual scene.

[1] Department of Neurobiology, Harvard Medical School, Boston, MA, USA. [2] Present address: Allen Institute for Brain Science, Seattle, WA, USA. [3] Present address: Department of Physiology, University of Tokyo School of Medicine, Tokyo, Japan. ✉email: clayr@alleninstitute.org

In trichromatic primates like macaques and humans, selectivity for color begins in the retina, where ganglion cells antagonistically combine signals from different parts of the visible spectrum to create cone-opponent representations[1]. The retina creates a "red/green" channel by comparing photon catches of long (L) and middle (M) wavelength-sensitive cones, and a "blue/yellow" channel from short (S) wavelength-sensitive cones and a combined L and M signal (L + M). This information is relayed to V1 in a highly specific pattern, with red/green inputs projecting to the middle cortical layer 4C and blue/yellow inputs terminating exclusively in superficial layers 2/3 and 4A[2].

How these chromatic signals are distributed and processed within V1 is less clear. The earliest electrophysiological studies[3–5] suggested that chromatic information is segregated within histochemical columns known as cytochrome oxidase (CO) blobs[6] and processed independently of spatial attributes before projecting to the ventral visual pathway. This idea of dedicated color modules was not supported by subsequent electrophysiological studies[7,8], contributing to a lingering lack of consensus about the existence of a color architecture[9,10]. However, the development of intrinsic signal imaging[11,12] rekindled the idea of chromatic maps. With their limited-resolution, bird's-eye view of cortex, these studies have consistently described small cortical patches, roughly associated with blobs, that respond preferentially to chromatic visual stimulation[13–15]. Here, we examined the fine-scale arrangement of these chromatic patches in macaque V1 with single-cell resolution using two-photon calcium imaging[16,17].

## Results

**Responses to spatially uniform chromatic stimuli**. We began by recording the patterns of activity evoked when an animal views full-field, uniform color (see Methods, Supplementary Fig. 1a). Cells that respond to unstructured stimuli are part of a broad class of chromatic cells with diverse receptive-field organization[3,8,18–20], including those preferring structured stimuli like color edges. However, multiple studies[7,21–23] have found that the early visual system's relative sensitivity to chromatic information increases as a stimulus approaches spatial uniformity, even if the optimal stimulus for a given color-selective cortical neuron is structured. Thus, we labeled neurons in layer 2/3 with calcium indicator dye (Fig. 1a) and asked whether flashed, uniform color fields stimulating specific cone classes reveal any fine-scale maps of chromatic preference.

When we plotted stimulus-evoked responses as $\Delta F$ maps (change in fluorescence between visual stimulation period and blank) we observed tight clusters of increased activity (Fig. 1b). The active cells in this field of view (FOV) carry hallmarks of low-level chromatic processing, including largely opponent signals from L and M cones when both are large enough to detect, along with strong, often dominant signals from short (S) wavelength-sensitive cones. Not only are active cells unambiguously clustered, but there exists substructure within these ensembles, a micromap of sign-specific cone preference (Fig. 1c; individual cells seen as bright spots). The +S (ON response) and −S (OFF response) subclusters are completely segregated, flanking a +L and −M group at center-left which, in turn, is distinct from a weaker +M and −L group at center-right. Also, while the +S and −S domains do not coincide, they do overlap to a small extent with the L/M subclusters, contributing "blue" input to cells primarily responsive to stimuli along the red/green axis (see Supplementary Fig. 2 for single-cell response time courses; Methods for color classes).

To verify that the clustering of cells that responded to uniform color is significantly different from random distributions, we extracted the spatial positions of all segmented cells in a given FOV and compared the observed clusters with those arising from randomly permuted cell positions. In the experiment illustrated

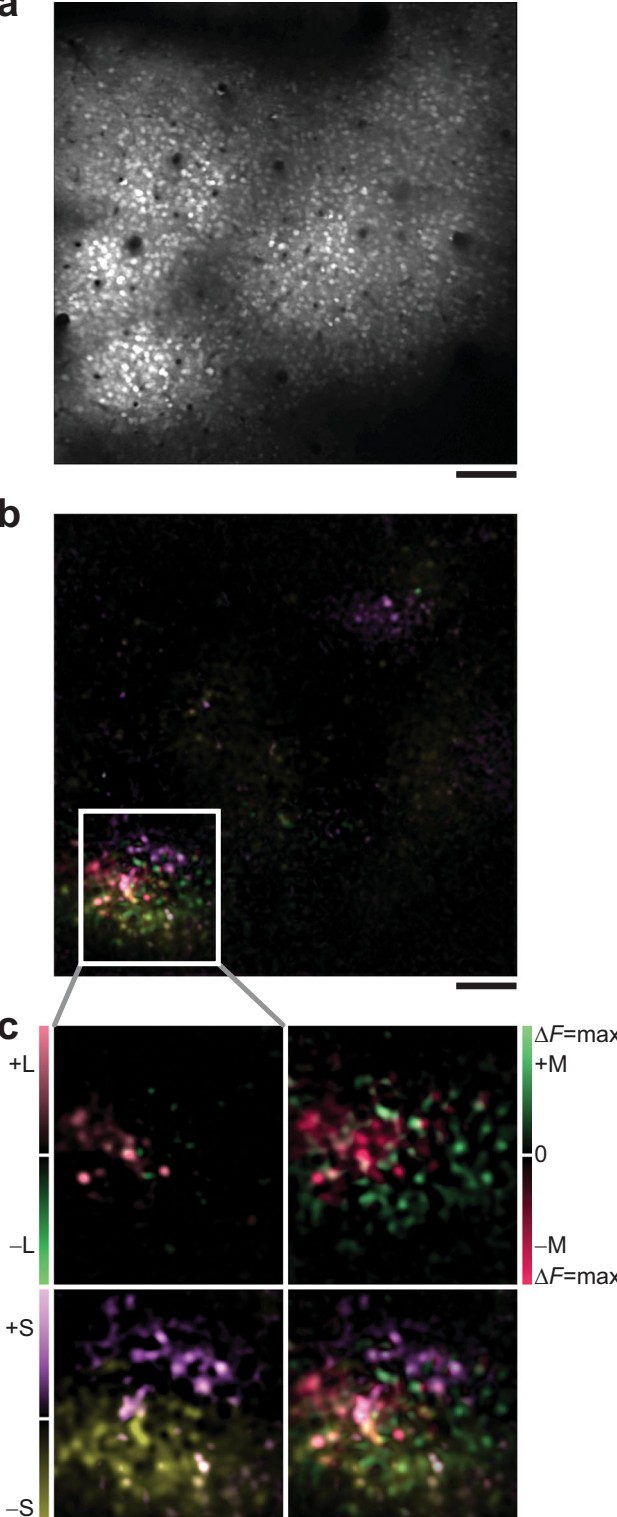

**Fig. 1 Clustering of cells into chromatic micromaps. a** Cells stained with calcium indicator, 215 μm depth. **b** Overlay of six single-condition $\Delta F$ maps obtained with spatially uniform stimuli, from field of view in **a**. Responses are to +/−L, +/−M, and +/−S conditions. **c** Magnification of cell cluster in **b**. Each panel has two color-coded $\Delta F$ maps, showing responses to sign-specific stimulation of a single cone type. Maps summed and rescaled at bottom-right (see Methods). Same color code in **b** and **c**, approximating color of stimulus. Scale bars, 100 μm.

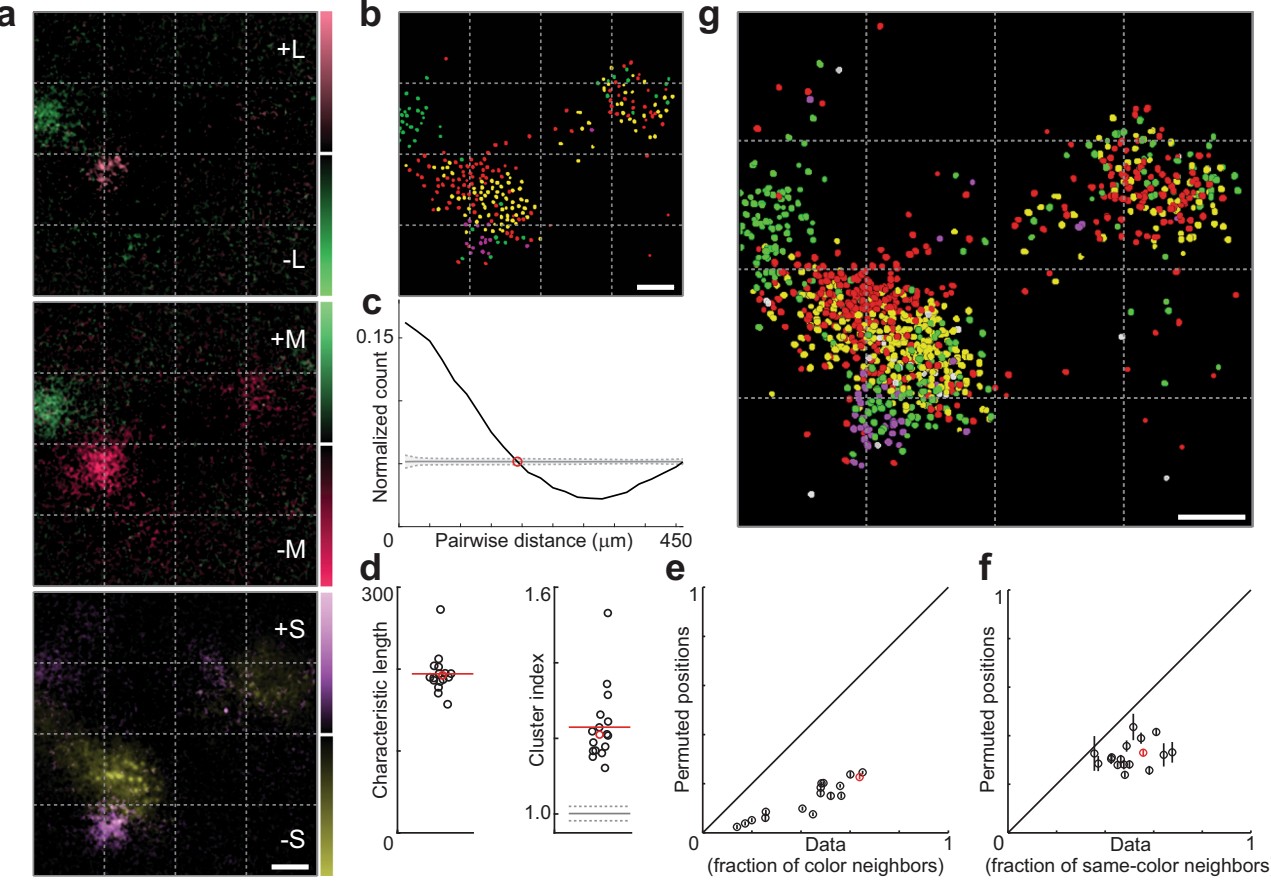

**Fig. 2 Substructure of micromaps. a** $\Delta F$ maps imaged at 230 μm depth. **b** Cell-based map derived from **a**. **c** Fraction of all possible cell pairs that are both responsive, as a function of pairwise distance (black line; data from FOV in **b**). Position-shuffled control in gray, dotted lines ±1 SD across permutations ($n = 1000$). Red circle marks characteristic length of observed data. Bin width, 20 μm. **d** Characteristic lengths for 17 FOVs (left). Clustering indices for same FOVs (right); gray line, mean clustering index for position-shuffled controls; dotted gray lines, ±1 SD ($n = 17,000$). Red lines, data means. **e** Comparison of fraction of responsive cells within 40 μm radius of responsive cells, for observed and permuted positions, same fields of view. **f** As **e**, except comparison of same-color neighbors. Error bars in **e**, **f** ±1 SD across permutations ($n = 1000$). Red markers in **d**–**f** represent field of view in **b**. **g** Overlay of cell maps taken from five different depths between 200 and 320 μm (988 responsive cells out of 5221). Scale bars, 100 μm (shared in **a**). Source data are provided as a Source Data file.

in Fig. 2a, a chromatic micromap with distinct L–M, M–L, +S, and –S subdomains forms an elongated cluster on the left side of the FOV. A cell-based response map (Fig. 2b) was generated by plotting colored symbols corresponding to the preferred stimulus of each significantly responsive cell (see Methods, Supplementary Fig. 2). We calculated the fraction of all possible cell pairs in which both cells were responsive, as a function of pairwise distance (Fig. 2c, black line). The position-shuffled control (gray line) is flat, whereas the unpermuted distribution of responsive pairs starts well above the shuffled line (more clustered), then dips below as the distance increases, until it finally approaches the shuffled line again. We defined the characteristic length to be the point where the measured distribution crosses the control. Plotting these values for 17 fields of view from four animals verifies that populations of cells responsive to uniform color had characteristic lengths clustered tightly around ~200 μm (Fig. 2d, left). Computing an index of clustering (Fig. 2d, right), used previously in both electrophysiological and imaging studies[24–26], showed that responsive cells were significantly more clustered than position-shuffled controls in these planes ($P = 2.9 \times 10^{-4}$, Wilcoxon signed-rank test; see Methods).

Chromatic micromaps were further analyzed using a likeness ratio, defined as the fraction of each responsive cell's nearest neighbors that also responded to uniform color. If clusters are

nonrandom, then the fraction should be lower in permuted data. This prediction was confirmed for all 17 fields of view (Fig. 2e). We used the same metric to characterize the subclusters themselves, with the likeness ratio defined as the fraction of each responsive cell's neighbors that belonged to the same color class. Responsive cell positions were shuffled only between responsive cells, and across our population, color classes formed nonrandom subclusters (Fig. 2f).

The FOV in Fig. 2b also illustrates the columnar organization[4,27,28] of chromatic micromaps. We found that full micromaps extended to form columns in upper layer 2/3, as shown by collapsing the data collected from five different depths into a single map (Fig. 2g; see Supplementary Fig. 3a–e for individual depths). The subdomains of the cell-based maps, particularly of the most responsive cluster to the left, recapitulate those seen in the original $\Delta F$ map (Fig. 2a), although the segregation is not absolute.

We examined the relationship between chromatic micromaps and histologically defined blobs (see Methods, Supplementary Fig. 4) and found that the two were highly correlated (example in Fig. 3a; representative maps from four animals in Supplementary Fig. 5). Each significantly responsive cell was assigned a number between 0 and 1, based on the magnitude of CO staining at the cell's location (one being the darkest pixels of each

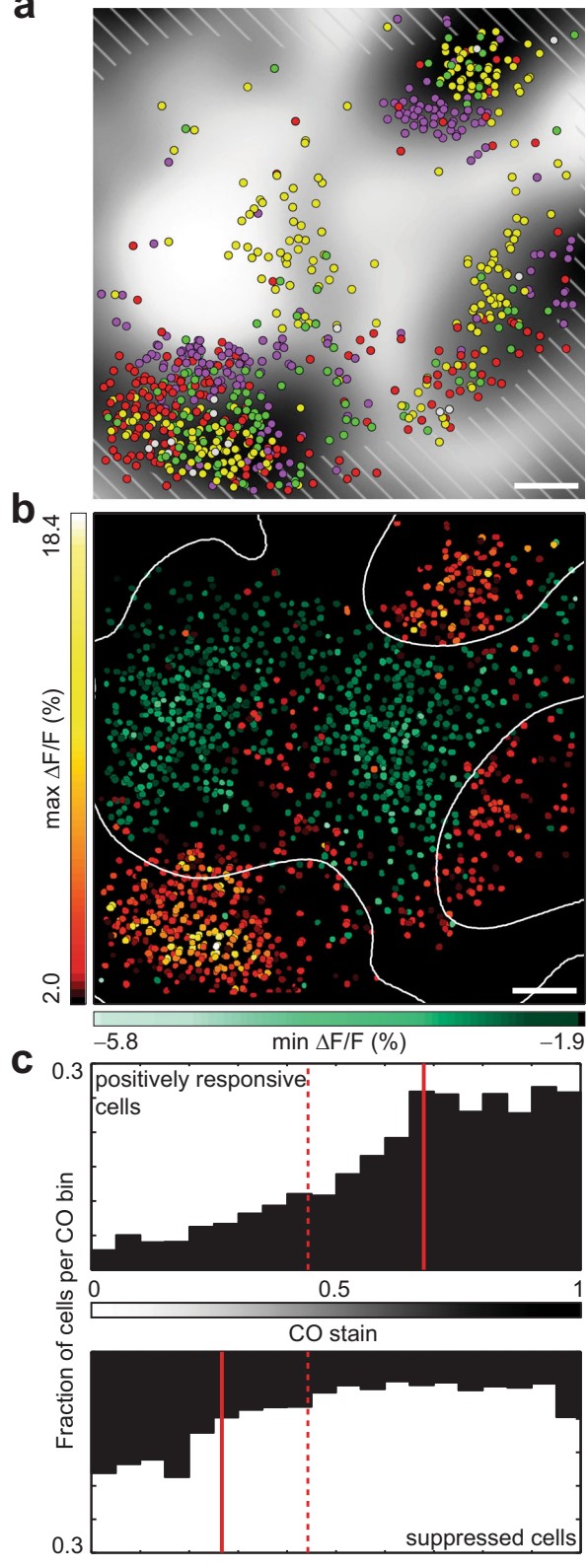

**Fig. 3 Relationship between micromaps and blobs. a** Cell-based map drawn on grayscale CO histology. Hatch marks signify lower quartile of indicator brightness. Data combined from 195 and 215 μm depths. **b** Maps showing strongest responses (max ΔF/F) of cells in **a**, and minimum responses (min ΔF/F) of suppressed cells from same fields of view. Contour lines show midpoint-gray level of CO histology, outlining blobs. **c** Relationship between response type and CO staining, population summary. Bars represent fraction of cells at given level of CO staining (bin width = 0.05) that responded significantly to spatially uniform stimuli, either positively (top panel; $n = 3365$ cells) or negatively (bottom; $n = 2242$). Solid red lines show median CO values for each response type. Dashed red line shows median CO value for all identified cells. Scale bar, 100 μm. Source data are provided as a Source Data file.

24,540; median CO = 0.44, IQR = 0.23–0.70) was highly significant ($P < 10^{-100}$, Wilcoxon rank-sum test).

Surprisingly, this spatial organization was critically dependent on the sign of the response. Identifying those cells that gave only significant negative responses (see Methods) shifted the pattern of activation away from strongly stained CO regions (Fig. 3b; time courses in Supplementary Fig. 2). The difference in median CO between these broadly suppressed cells (Fig. 3c, lower panel; $n = 2242$; median CO = 0.27, IQR = 0.14–0.52) and the entire population was also highly significant ($P = 4.3 \times 10^{-89}$, Wilcoxon rank-sum test).

**Responses to spatially structured chromatic stimuli.** We next asked whether the envelope of activity containing micromaps and suppression was linked to the spatial uniformity of our stimuli, since blobs are more responsive on average to lower spatial frequencies than interblobs[29,30] and since many color cells respond best to structured stimuli[3,8,18–20]. In 12 fields of view, we ran a second set of experiments in which the same chromatic stimulus conditions were presented in the form of oriented, drifting bars (Supplementary Fig. 1b). One of the clearest effects of adding structure was an expansion of responsive clusters and an increase in peak activity (example in Fig. 4a). Neurons responding to bars were significantly clustered (Fig. 4b; $P = 4.9 \times 10^{-4}$, Wilcoxon signed-rank test), and there were more responsive cells on average to bars than to uniform conditions (Fig. 4c; paired $t_{1,241} = 3.03$, $P = 0.011$). Many cells responded to both structured and unstructured color (1242 coactive cells; 1940 uniform-only; 3705 bars-only; out of 14,576 segmented), with coactive cells responding more robustly on average to structured color (Fig. 4d; paired $t_{1,241} = 19.4$, $P = 1.2 \times 10^{-73}$). Out of 1242 coactive cells, 929 showed increased peak response to drifting bars. We observed no significant difference in the distribution of CO values of these cells compared with those that showed decreased peak response ($P = 0.50$, two-sample Kolmogorov–Smirnov test), and median CO values were nearly identical (0.67 and 0.69, respectively). Notably, general suppression disappeared completely with structured stimuli (1660 suppressed cells to uniform stimuli; nine to bars).

Color tuning was also highly sensitive to stimulus spatial structure. Spatially uniform conditions evoked responses from cells in chromatic micromaps (e.g., Fig. 2a; time courses in Supplementary Fig. 2) similar to those in the dorsal lateral geniculate nucleus, as can be seen across the population in the distributions of responses to each color condition (Supplementary Fig. 6a) as well as in plots of overall tuning incorporating contributions from both red/green and blue/yellow color axes (Supplementary Fig. 7c, d and g, h; see Methods). There was some inter-axis mixing consistent with previous work[18], particularly blue/yellow contributions to red/green cell classes. However, responses to uniform stimuli were relatively pure in the sense of

grayscale, histogram-equalized blob map). The resulting distribution of CO values showed a pronounced tendency for positively responsive cells to reside in areas of darker cytochrome staining (Fig. 3c, upper panel; $n = 3365$; median CO = 0.68, IQR = 0.46–0.82), and the difference in median CO between positively responsive cells and the entire population of segmented cells ($n =$

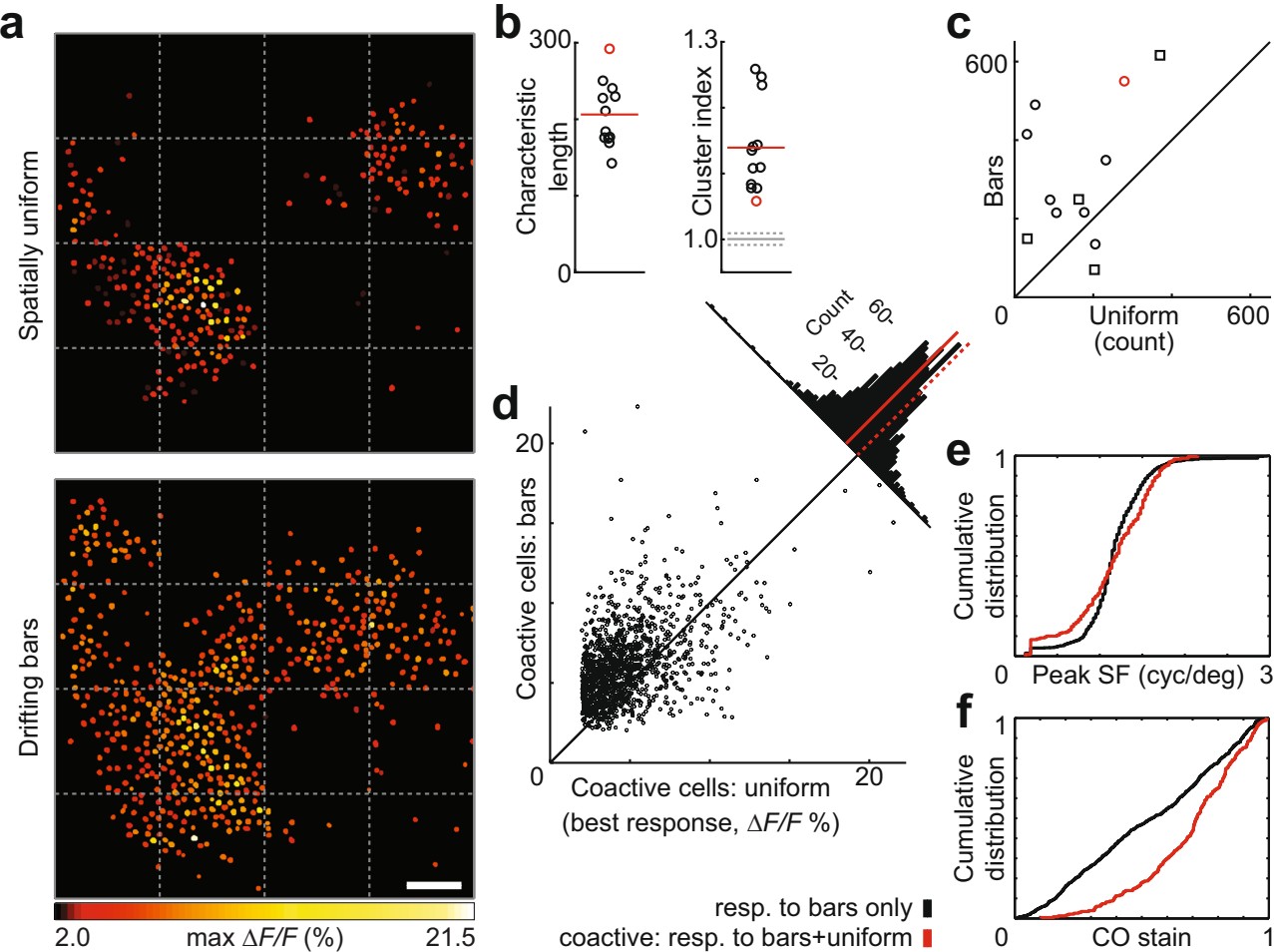

**Fig. 4 Responses to spatially structured vs unstructured stimuli. a** Upper: cell-based map of best response across all spatially uniform color conditions. Lower: best response of same cells to same color conditions presented as drifting bars (across four directions of motion). **b** Characteristic lengths for 12 FOVs with drifting bar stimuli (left). Clustering indices for same FOVs (right); gray line, mean clustering index for position-shuffled controls; dotted gray lines, ±1 SD (n = 12,000). Red lines, data means. **c** Number of responsive cells in same FOVs to uniform and bar stimuli. FOVs tested with four directions of motion are marked with circles, one direction of motion with squares. Red markers in **b**, **c** represent FOV in **a**. **d** Comparison of best responses to uniform and bar stimuli, same FOVs. Dashed red line marks zero difference, solid line marks mean (1.4%). **e** Cumulative distribution of peak achromatic spatial frequency preference for cells responsive only to chromatic bars (black line) or to both bars and uniform stimuli (red line). **f** Cumulative distribution of CO staining intensity for same set of cells. Scale bar, 100 μm (shared in **a**). Source data are provided as a Source Data file.

minimal intra-axis contributions, defined as responses to conditions from the opposite or sign-reversed end of a class's dominant axis, like green (+M, M–L, –L) responses from cells in the red (–M, L–M, +L) class (Supplementary Fig. 7).

This purity of color tuning to uniform stimuli was sharpened by surprising class-specific suppression to individual color conditions (Supplementary Fig. 6a), distinct from the general suppression to all conditions shown in Fig. 3. The distributions of responses to intra-axis (green) conditions were significantly suppressed in red cells, as were most red responses in green cells. The distributions of responses to all six red and green conditions were significantly suppressed in the blue (+S, –L–M) class. And at the extreme of inter- and intra-axis suppression, seen in the yellow (–S) cell class, the median response to every condition except –S was below baseline. All classes showed suppressed responses to the luminance condition (L+M), except for the 16 cells (1.3% of 1242 coactive cells) making up the luminance class.

With drifting bars, class-specific suppression disappeared. Median responses significantly increased to almost every condition in every cell class (Supplementary Fig. 6b), resulting in significantly more inter- and intra-axis mixing (Supplementary

Figs. 7 and 8a). Responses to red/green conditions increased far more than to blue/yellow across all classes (Supplementary Fig. 6d), eroding the clear distinctions between classes seen with uniform stimuli (compare Supplementary Fig. 7d, h and l, p). Using an alternative, vector-based classification of color tuning (since the best-response classification used with uniform stimuli does not capture intra- and inter-axis mixing; see Methods and Supplementary Table 2, right columns), we found that this unbalanced gain between the two color axes caused a redistribution of cells classified as blue/yellow to red/green, particularly away from the yellow (–S) class, which saw the smallest increase in response to structured stimuli. Even so, the chromatic maps defined with uniform stimuli were maintained with bars (Supplementary Fig. 9; single-condition maps in Supplementary Fig. 9a, b from same FOV as Fig. 4a), but each individual subdomain underwent an expansion as well, creating more overlap of previously distinct subdomains and new hotspots (Supplementary Fig. 9e; example single-cell time courses in Supplementary Fig. 9f).

Blue/yellow cells, which constituted 55.6% of coactive cells classified with unstructured conditions, fell to 22.9% of coactive cells

with structured stimuli (Supplementary Table 2), the latter figure more comparable to an early electrophysiological survey of color cells in well-delineated blobs[3]. In general, the observation that color tuning strongly depends on the spatial structure of the stimulus is broadly consistent with earlier electrophysiological studies that examined this question[7,22], with exceptions[31]. Crucially, however, the class-specific suppression results suggest that the geniculate-like responses seen with uniform stimuli are not the result of cells receiving input from, and functioning as cortical relays of, single geniculate pathways. Instead, there is substantial convergence of red/green and blue/yellow inputs to many of these micromap cells, and intra-cortical mechanisms sensitive to the spatial content of the stimuli shape the overall color tuning.

The receptive-field structure of neurons in chromatic micromaps remains an open question. Some were almost certainly Type II cells[3], which respond best to uniform color, but the increased activity to bars (Fig. 4d) and marked shift in color tuning with bars strongly suggests a significant fraction were neurons that signal color edges, such as double-opponent cells[3,18] or oriented color cells[8,19]. Double-opponent cells are not necessarily silent to unstructured stimulation, as their centers are often stronger than their surrounds[32]. These classes of cells, or some variation of them, could contribute to micromaps obtained with uniform stimuli via the dominant subfield of their receptive fields.

In a subset of experiments, we measured peak spatial frequency of the two populations that responded to structured color (bars-only and coactive cells) using achromatic, drifting gratings (Fig. 4e). The distributions, while significantly different ($n_{bars} = 788$, $n_{coactive} = 230$; $P = 0.0028$, two-sample Kolmogorov–Smirnov test), were overlapping with no significant difference in medians ($P = 0.61$, Wilcoxon rank-sum test). Thus, cells that responded to uniform color stimuli were not limited to the lowest preferred frequencies, again arguing against a largely Type II population. The groups diverged most clearly with respect to CO staining (Fig. 4f; $P = 4.0 \times 10^{-15}$, two-sample Kolmogorov–Smirnov test; same cells as in Fig. 4e). Cells responding to both uniform and bar stimuli clustered near blobs, but bars-only cells were distributed evenly between blobs and interblobs. We observed small or no significant differences in color tuning metrics between coactive cells in blobs and interblobs, or between coactive cells and either bars-only or uniform-only cells (Supplementary Fig. 8). These were largely overlapping distributions.

Finally, as a bridge to previous work, we also ran cone-isolating gratings (Supplementary Fig. 1c), commonly used in color studies[2,7,19,20,23,28] but avoided as primary stimuli for this study because they include both positive and negative contrasts. Although the gratings and bars had different chromatic and spatial characteristics, responses to gratings showed the same expansion and overlap of individual color subdomains and gave virtually identical maps to those obtained with sign-specific bars (Supplementary Fig. 10). There was also considerable mixing of L/M and S inputs at subdomain borders (Supplementary Fig. 10b), creating off-axis color responses. These roughly segregated cone maps are entirely consistent with recent two-photon work[33] examining hue maps in V1 with drifting grating stimuli.

Taken together, the structured stimulus experiments suggest that cells within chromatic micromaps are heterogeneous in the spatial organization of their chromatic receptive fields. Given the existence of different color cell types in upper layers of V1, there is very likely richer organization within the architecture of micromaps, whose description will require detailed characterization of spatiotemporal receptive fields.

## Discussion

Interpreting these results within the context of known circuitry suggests a more general hypothesis about the functional design of V1. Parvocellular neurons in the lateral geniculate nucleus, which provide the red/green input to V1, act as low-pass spatial filters for color information but are bandpass filters for achromatic or luminance stimuli[21], switching between the dual roles of chromatic and high-acuity signaling based on the spatial frequency of the stimulus. Unlike blue/yellow koniocellular geniculate afferents, which target upper layers directly[2] and are part of a system with lower acuity and less spatial antagonism, parvocellular signals arrive at the deeper input layer 4Cβ[2], where individual cells send axon collaterals to both blobs and interblobs in layer 2/3[10,34]. We hypothesize that the significance of this parallel input to blobs and interblobs may lie in the way cortex processes color and form. Sculpted by full-field suppression, the representation is restricted to blobs when the dominant stimulus feature is spatially uniform color, and within these blobs are micromaps containing ensembles that represent the convergence of both red/green parvocellular and blue/yellow koniocellular signals. With spatially structured stimuli the interblobs are no longer silent, so parvocellular contributions to form vision are incorporated into the representation, as are the chromatic boundary-detection contributions of blob cells. Thus, V1 may perform as a switch, shifting the ensembles it uses to represent spatial and chromatic information across CO compartments.

## Methods

**Animal preparation and surgery**. We used four adult macaque monkeys (*Macaca mulatta*, three males and one female, 8–12 years old, 8–18 kg) in this study. Each animal was initially anesthetized with ketamine HCl (10–20 mg kg$^{-1}$) and xylazine (0.5 mg kg$^{-1}$), tracheotomized, and placed in a standard stereotaxic apparatus. Anesthesia was maintained with sufentanil citrate (4–15 µg kg$^{-1}$ hr$^{-1}$, i.v.) during surgery and recording. Dexamethasone (0.5 mg kg$^{-1}$, i.m.) was given every 24–48 hours to reduce brain swelling. A stainless steel chamber was implanted on the skull with dental cement, and a craniotomy (~3 × 3 mm) and durotomy were performed posterior to lunate sulcus above the operculum, exposing the parafoveal representation of V1. Cortex was covered with agarose (2–3% in artificial cerebrospinal fluid). The animal was paralyzed (pancuronium bromide, 0.1–0.2 mg kg$^{-1}$ hr$^{-1}$, i.v.) after the completion of all initial surgery. Eyes were dilated with 1% atropine and corneas protected with gas-permeable contact lenses. Optimal refraction was achieved using a streak retinoscope and external lenses. Body temperature, electroencephalography, electrocardiogram, expired $CO_2$, and heart rate was monitored continuously to judge the animal's health and maintain proper anesthesia levels. All surgical and experimental procedures were in accordance with National Institutes of Health and United States Department of Agriculture guidelines and were approved by the Harvard Medical Area Standing Committee on Animals.

**Two-photon calcium imaging**. Cortex was bulk-loaded[16,17] with calcium indicator. A patch pipette (tip broken to a diameter of 2.5–4 µm) was filled with 1–2 mM Oregon Green 488 BAPTA-1 AM (OGB-1 AM) dissolved in 8% dimethylsulfoxide, 2% pluronic acid, and 40 µM Sulforhodamine 101 (all from Molecular Probes). The pipette tip was inserted to a depth of 250–300 µm below the pial surface, where dye solution was pressure-injected (5–25 psi, 1 s pulses every 10 s for 10–15 min). The injection procedure was repeated 2–3 times, with pipette insertions laterally spaced ~500 µm apart. Multiple overlapping injections resulted in labeled fields spanning >1000 µm across, labeling up to ~2000 neurons for simultaneous imaging. Since blobs are spaced ~350–550 µm apart[6], these imaged areas were sufficient to monitor cells in multiple blobs, along with large expanses of interblobs.

The craniotomy was sealed with a glass coverslip, and calcium fluorescence in neurons was monitored with a Sutter MOM two-photon microscope coupled to a Chameleon Ultra (Coherent Systems) mode-locked Ti:sapphire laser. Data acquisition and microscope control was achieved with custom code and the MPScope software package[35]. Excitation light (920 nm) was focused by a 16x water immersion objective (Nikon, 0.80 NA). A square region 765 µm on a side was imaged at 512 × 512 pixels at 2 frames s$^{-1}$. The average power delivered to the brain was <70 mW. Images from multiple depths, spaced ~20 µm apart, were obtained for most stimulus protocols.

**Visual stimuli**. All stimuli were presented on a CRT monitor at 75 Hz refresh rate, driven by a custom stimulus package based on DirectX on a PC graphics card. The additivity of the red, green, and blue guns was confirmed, and gun intensities linearized with respect to frame buffer value. Cone-specific directions were calculated from human 10° cone fundamentals[36] and the measured spectral power distributions of the monitor phosphors[37] obtained with a PhotoResearch PR

650 spectrophotometer. Stimulus presentation was not synchronized with data acquisition.

The primary stimulus set consisted of spatially uniform flashes stimulating specific cone classes (ten conditions: L+M, +M, M–L, –L, –L–M, –M, L–M, +L, +S, –S; Supplementary Fig. 1a). Each condition was flashed at a 2–3 Hz rate for 3 s from a neutral gray background (linearized guns at half-maximal intensity; mean luminance ~37 cd m$^{-2}$ and CIE color coordinates $x = 0.28$, $y = 0.31$). Conditions were presented at the maximum cone contrasts achievable by our monitor. Michelson contrasts $[(I—I_0)/(I + I_0)$, where $I$ is the intensity of the chromatic stimulus for a given cone type, and $I_0$ is the intensity of the neutral gray], were 0.20, 0.07, 0.04, 0.08, 0.36, 0.09, 0.04, 0.07, 0.33, 0.80, respectively. Conditions stimulating two cone classes together had matched contrasts for each cone class, represented by the given value. In a second stimulus set (Supplementary Fig. 1b), we presented the same ten color conditions as drifting bars (2 Hz temporal frequency, 20–30% spatial duty cycle, barwidth of 0.7–1.5°). A third stimulus set (Supplementary Fig. 1c) consisted of cone-specific, sinusoidal gratings (five conditions: L, M, S, L+M, L–M; drifting at 3 Hz temporal frequency, spatial period of 1.25°), with grating peak and trough modulating about neutral gray. Cone contrasts for L (M, S) were 0.15 (0.18, 0.89), and for the L±M conditions, L and M were matched at 0.08. A fourth stimulus set consisted of achromatic gratings at full contrast with logarithmically spaced spatial frequencies (0.125–2.9 cycles per degree, 4 Hz temporal frequency). For bars and gratings, responses to different directions of motion (up to four) were collected as separate runs. Each condition was followed by a blank of the same duration (3 s for uniform, 3–6 s for bars and gratings), and all stimuli were presented sequentially and repeated 10–20 times. Conditions were not randomized, but the long duration of neutral gray between each condition likely mitigates adaptation concerns, particularly with respect to general micromap structure and how the spatial content of stimuli affects the functional organization.

Stimuli were displayed full field with the monitor ~40 cm from the animal's eyes. Stimuli covered ~23 × 30 visual degrees. We did not measure receptive-field sizes of recorded neurons, but recording locations corresponded to ~6–8 degrees eccentricity and receptive-field centers ~2 degrees in diameter[38,39]. All data shown came from binocular stimulation, but the contralateral and ipsilateral eyes were also stimulated alone to determine ocular dominance and confirm that color maps were stable between monocular and binocular presentation.

**Histological reconstruction**. At the termination of experiments, the animal was deeply anesthetized and transcardially perfused (phosphate-buffered saline, followed by 4% paraformaldehyde, 10 and 20% sucrose). Cortex was blocked, sunk in 30% sucrose, and flattened. A thick (~200 µm) tangential section of the imaged cortex was cut on a freezing microtome to include all traces of surface vasculature and upper layer 2/3, followed by a series of 50 µm sections. Sections were stained for CO, then cleared[40] in a solution of one part benzyl alcohol in two parts benzyl benzoate (Sigma).

Reference two-photon images were generated by averaging >100 frames of motion-corrected data for each plane (e.g., Fig. 1a), and alignment between two-photon images, in vivo photographs, and images of stained tissue was achieved using surface and radial vasculature. No transformations beyond isotropic scaling and rotation were necessary for alignment. Blob images (cropped substantially larger than the two-photon images) were processed to remove radial vessel artifacts (Photoshop), converted to grayscale, filtered (FFT bandpass filter, 43–430 µm), smoothed (Gaussian, $\sigma = 43$ µm), and histogram-equalized, all in ImageJ (National Institutes of Health). See Supplementary Fig. 4 for end-to-end illustration.

**Data analysis**. Functional data were analyzed in Matlab (MathWorks). Time-lapse two-photon images were realigned using a procedure based on TurboReg[41], which corrects for tangential movement of the FOV by maximizing the correlation between frames. Cells were segmented by template matching, and single-cell fluorescence time courses were extracted by summing pixel values within cell contours. Slow drifts in baseline fluorescence were removed with a low-cut filter (cutoff: duration of two complete stimulus repeats). Visually responsive cells were defined by analysis of variance across blank and all conditions ($P < 0.01$), with further selection criteria of maximum $\Delta F/F$ (change in fluorescence normalized by the baseline fluorescence) >2.0% for positively driven cells, or maximum $\Delta F/F < 0$ in response to all 10 color conditions for suppressed cells.

We used two types of maps to show the spatial organization of responses within a FOV: pixel-based (e.g., Fig. 2a) and cell-based (e.g., Fig. 2b). Each pixel-based map is a $\Delta F$ image, defined as blank image subtracted from the condition image, where blank image is the average frame immediately preceding all stimulus conditions. Condition images were averaged over stimulus repetitions and maps were spatially smoothed (Gaussian, $\sigma = 2.8$ µm). We used $\Delta F$ instead of $\Delta F/F$ responses for clarity of presentation, as the denominator in $\Delta F/F$ maps can introduce noise from areas of low resting fluorescence, like vascular shadows. Also, $\Delta F$ maps are more appropriate for comparing signals between cell bodies and neuropil, making it easier to visualize the puncta of cell bodies in higher magnification images (e.g., Fig. 1c). To generate pixel-based $\Delta F$ maps combining multiple cone conditions, each single-condition $\Delta F$ map was first normalized by cone contrast, and then the +L, –L, +M, and –M maps were scaled as a group to

maximize dynamic range for clarity, as were +S and –S maps. All maps of derived variables (such as preferred color group or best response across multiple orientation runs) are visualized as cell-based maps using $\Delta F/F$ responses, with icons marking the position of each cell in the FOV. For cell-based analyses comparing data from multiple depths, two-photon images were realigned using radial vessels to correct for tilt.

OGB-1 AM responses are known to follow a saturating nonlinearity[42,43], which can complicate the interpretation of contrast-normalized responses. However, sublinear saturation effects become prominent only when OGB-1 AM signals are greater than approximately half the maximum possible signal, which can be 300% $\Delta F/F$ or more in many preparations[42]. The responses in this study were much smaller (rarely above 20% $\Delta F/F$), placing these responses well below the onset of significant saturating nonlinearities.

Cells were assigned to color classes based on their preferred spatially uniform condition, using positive $\Delta F/F$ responses normalized by each condition's cone contrast (see Supplementary Fig. 2). Suppressed responses were not normalized for any analyses. There were five classes: red cells responded best to +L, L–M, or –M; green to +M, M–L, or –L; blue to +S or –L–M; yellow to –S; and luminance to L+M. These designations also applied to the stimulus conditions themselves (e.g., +L, L–M, –M are referred to collectively as red conditions). Class names reflect the approximate color of their conditions. For analyses comparing responses to spatially uniform and drifting bar stimuli (Supplementary Figs. 6–8), initial class designations based on uniform stimuli were carried through to bar analyses, keeping comparisons within the same cell populations. Discrete classes based on the best response to uniform conditions simplified the interpretation of cell-based maps, both visually and in analyses like subclustering within micromaps (Fig. 2f). They were justified by comparing with an alternative, vector-based quantification of color tuning, in which we plotted responses on a plane defined by orthogonal red/green and blue/yellow axes (red/green coordinate from Supplementary Equation 2, blue/yellow from Supplementary Equation 3, in Supplementary Fig. 7). Each cell's overall color tuning in this plane was given by the angle θ between the positive $x$ axis and the vector from the origin to the cell's location in the plot. For uniform stimuli, the two classification methods largely resulted in the same populations (Supplementary Fig. 7d, h; see also Supplementary Table 2). However, the preferred-condition method was not appropriate for classifying cells based on responses to bars, independent of uniform stimuli, since intra- and inter-axis mixing with bars precluded unambiguous assignment to a single color class by the single best response (see Supplementary Figs. 6 and 7). To independently classify cells based on structured stimuli, we used the vector analysis method described above (Supplementary Table 2).

For analyses related to CO, numerical CO values for cells were calculated by aligning each two-photon FOV with its grayscale blob image and averaging the pixel values of the map found within the contour of each segmented cell.

An index of clustering[24–26] (Figs. 2d and 4b) was determined for each FOV by first calculating the median absolute pairwise distance between responsive cells across all pairs located within 200 µm (approximately the mean characteristic length overall FOVs), and then doing the same calculation for each of 1000 position-shuffled controls. The clustering index for each FOV was then defined as the median of the median pairwise distances of all position-shuffled controls (the grand median) divided by the median pairwise distance of the observed data. If this ratio is 1, the observed clustering is taken to be similar to a random distribution. The more the ratio exceeds 1, the greater the clustering.

Modified kernel density estimation (Supplementary Fig. 9a, b) was done using cells in the upper quartile of response strength in each single-condition map. We used a two-dimensional Gaussian kernel with amplitude proportional to cell response:

$$K(x, y) = R \cdot e^{-\frac{((x-x_0)^2 + (y-y_0)^2)}{2\sigma^2}} \tag{1}$$

where $R$ is the $\Delta F/F$ response of the cell; $x_o$ and $y_o$ are the coordinates of the cell's position in the FOV; and $\sigma$ is the standard deviation of the Gaussian kernel ($\sigma = 30$ pixels). KDE maps were obtained by summing $K$ over the population of cells in each FOV. Peaks were determined by comparing neighboring pixels, as implemented in the findpeaks2 algorithm in Matlab. Two-dimensional correlation coefficients (Supplementary Fig. 9e) were computed with the corr2 algorithm in Matlab.

For spatial frequency experiments, peak spatial frequency was extracted from a fit of each cell's $\Delta F/F$ responses using a difference of Gaussians model[31] of the form:

$$F(f) = F_0 + K_c \cdot e^{-\left(\frac{f-\mu_c}{2\sigma_c}\right)^2} - K_s \cdot e^{-\left(\frac{f-\mu_s}{2\sigma_s}\right)^2} \tag{2}$$

where $f$ is spatial frequency; $F_0$ is baseline; and $K$, $\sigma$, and $\mu$ define the Gaussian components (center and surround denoted by subscripts $c$ and $s$, respectively).

**Statistics and reproducibility**. All statistical tests are described in the text or figure legends. Unless otherwise noted, tests are two-sided and statistical significance is defined as $P < 0.01$. No statistical methods were used to determine sample size, but the numbers of animals and FOVs used in this study are consistent with other published work. Every FOV analyzed for color maps is listed in

Supplementary Table 1, and none were excluded except for failed labeling with calcium indicator. Representative micrographs (e.g., color clusters in Fig. 1) highlight observations consistent across all FOVs.

**Reporting summary**. Further information on research design is available in the Nature Research Reporting Summary linked to this article.

## Data availability
Source data for Fig. 2c–f; 3c; 4b–f; and Supplementary Figs. 6a–c; 7a, c, e, g, i, k, m, o; 8a–c; and 9c–e are provided with this paper. Other data that support the findings of this study are available from the corresponding author upon reasonable request. Source data are provided with this paper.

## Code availability
Analyses were performed using MATLAB and ImageJ. Code available from the corresponding author upon reasonable request.

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

## Acknowledgements
We thank A. Vagodny for surgical assistance; S. Yurgenson for technical assistance and programming; D. Troilo for providing New World primates used in preliminary experiments; R. Shapley, B. Conway, and M. Hawken for critical reading of the manuscript; and the New England Primate Research Center. Research reported in this publication was supported by the National Eye Institute of the National Institutes of Health under Award Numbers R01EY010115 and F32EY016287. The content is solely the responsibility of the authors and does not necessarily represent the official views of the National Institutes of Health. K.O. was additionally supported by Brain/MINDS-AMED, WPI-IRCN, Institute for AI and Beyond, and KAKENHI (19H05642, 20H05917).

## Author contributions
S.C., K.O., and R.C.R. designed the experiments; S.C. and K.O. collected and analyzed data; S.C., K.O., and R.C.R. wrote the paper.

## Competing interests
The authors declare no competing interests.
