## [Peer Review File · Nature Communications]

Reviewers' Comments:

Reviewer #1:

Remarks to the Author:

In this paper, Chatterjee and colleagues use two-photon calcium imaging to investigate the micro-organization of color tuning in superficial layers of monkey V1. This is a unique study that will contribute important insights to our understanding of the functional organization of V1. The paper will likely be of broad interest for the general neuroscience community.

I have a number of minor suggestions, mostly regarding the data analysis:

- For most of the paper, two-photon data are analyzed as dF/F , with the exception of the pixel-based maps. If there is a reason for this choice, it would be useful to discuss it; otherwise the authors may wish to plot dF instead of dF/F . In addition, the normalization of responses across cone contrast appears problematic, as OGB responses are known to follow a saturating nonlinearity. The effects of this nonlinearity on the conclusions should be discussed.
- It would strengthen the results if the authors could show maps for more of the imaging regions included in the data analyses.
- Currently, the cell-based analysis may overemphasize distinctions between cells, as each cell is assigned to one color category only based on its preferred response. It might be better to quantify color preference with a vector in color space that represents not only the best response, but overall color tuning.
- Similarly, it would be useful to include an analysis of potential changes in color preferences across stimulus types.
- While the increase in the number of responsive neurons for bars versus uniform stimuli is obvious, there currently is no statistical test for it.
- The text states that color modules overlap more strongly for bars versus uniform stimuli, but does not provide quantification for this statement.
- The characteristic length measure appears problematic when multiple clusters are present in the same imaging region. Neuron pairs spanning clusters will contribute long inter-pair distances, which will form a second peak in the distance histogram. The relative height of the peaks will depend on the size of the clusters. As the characteristic length is determined at half peak height, it can be affected by the presence and relative size of the other peaks. In addition, it captures the differences in map structures induced by bars versus uniform stimuli rather indirectly: The increase in responsive neurons with bar stimuli will likely result in filling in the 'middle distance' range of the distance histogram, rather than changing the initial part of it. The only reason for an increased characteristic length is then that this filling in is strong enough to move the location of the first peak, rather than demonstrating the overall larger region. It might be good to add a second metric that more directly captures the compactness of the responsive regions.
- Suppression of responses seems to occur for all colors based on the example cells. Is that generally the case?
- To be able to correctly interpret the result shown in Figure 1G, either maps for each imaging depth should be shown, or the number of cells included per depth need to be stated (otherwise it cannot be ruled out that the result is simply due to the majority of cells coming from one imaging depth only).
- Clarification for methods: How large was the stimulus in visual degrees, and how large were the estimated receptive fields of the recorded neurons? At which eccentricities were the experiments done approximately?
- Clarification for methods: Was the sequence of stimulus conditions randomized? If not, are there concerns regarding adaptation?
- Abstract: The abstract claims a columnar organization of color tuning. Two-photon imaging is limited to upper layer 2/3 in macaques, and can therefore not demonstrate a full columnar organization. It would be better to either clearly state the limit in the abstract, or to rephrase columnar to clustered/patchy organization.

Reviewer #2:

Remarks to the Author:

This paper describes a 2-photon calcium imaging study that addresses a long-standing issue in color neurophysiology: Are the spatially lowpass, chromatically opponent neurons of the macaque primary visual cortex enriched in the cytochrome oxidase-enriched blobs? The answer appears to be an unequivocal "yes". I enjoyed reading and learning from this paper.

Major comments:

Which neurons contribute to color vision is an unresolved issue. For this reason, I find the authors' use of the word "chromatic" (e.g. in the context of "chromatic cells" and "chromatic maps") confusing. My reading of the paper suggests that a luminance-tuned cell that doesn't contribute to color vision but responds well to uniform stimuli would have been included in the "fine-scale map of chromatic preference". Is this so? Were such low-pass, non-opponent cells observed? An analysis of cone weights would be useful. Do non-opponent neurons cluster spatially in the cortex similarly to opponent neurons? Among cone-opponent neurons, is there a tendency for the sign of the S-cone weight to match the sign of the M-cone weight, as electrophysiological experiments have found? Does the degree of cone-opponency differ between neurons in the blobs and the interblobs when probed with bars or gratings?

How much of the spatial structure in the photomicrographs is due to neural activation and how much is due to loading of the dye? The Methods mention that the dye spread over ~ 1 mm, and I assume that the dye penetration was roughly uniform over this area, but an analysis of the spread of the dye relative to the size of the activated cell clusters, if possible, would be welcome.

Minor comments:

Throughout the paper, stimuli are referred to as "cone-isolating" but four of them—L-M, L+M, -L-M, and M-L—do not isolate a single cone type.

The final sentence of the abstract and of the discussion refer to "dynamic switching". Temporal changes in V1 activity are not addressed in this study, so it seems strange to conclude the paper with a speculation about them. I realize that different spatio-chromatic patterns of light entering the eye produce different patterns of cortical activity, and that such patterns change over time, but I do not see how the results of this study add to this picture. Along the same lines, I am unsure what the word "robust" in the last sentence of the abstract is intended to mean.

"The statistics of natural scenes also support a preferentially low-spatial-frequency color system." I do not see how the statistics of natural images motivate a color system that is tuned for lower spatial frequencies than the luminance system.

Referring to the L, M, and S cones as "red", "green" and "blue", even parenthetically, is counterproductive.

Cell-based maps: Please state the number of cell classes (I believe the answer is 4) and clarify how the categorization was done. I think most of the relevant information is in the legend to Extended Data Figure 2, but this material is sufficiently important that it should be moved to the Methods or to one of the figure legends in the main text.

"0.06%; non-unique ..." I don't understand this.

Why are t-tests used for some comparisons and Wilcoxon tests for others?

How was the image in Figure 1A obtained? Does it reveal where the dye-loaded cells were located,

independently of their stimulus preferences? Does it show the locations of all responsive cells? Why there are fewer visible cells in Figure 1b?

Figure 2. I am confused about the meaning of the gray curves. The integral of the gray curves and the black histogram are similar, suggesting that similar numbers of cells were used to create each of them. Yet, the text says that the curves were obtained by "repeatedly shuffling the positions of all identified cells within the field of view". Also, a single gray curve follows the black histogram closely. This presumably is not a result of random permutation, but this should be made clear.

Was the 2 Hz presentation of the spatially uniform and bar stimuli synchronized to the 2 Hz data acquisition?

Extended Data Figure 5 legend: "significantly responding to > 8 conditions". "Stimulus condition" could mean spatial structure, but I do not think that it does here. Does this refer to 8 of the 10 basic stimulus conditions (+L, +M, -L, -M, L-M, M-L, -L-M, L+M, -S, +S)?

We thank the reviewers for supporting the work and their tremendously helpful suggestions. We performed a series of new analyses to address their comments, and we revised the manuscript to incorporate every point made. Since the review occasioned a large number of changes, we quote only the most substantial in our responses below. All additions to the manuscript, as well as our responses to the reviewers, are indicated in red.

Summary of major changes:

1. New analyses of cell clustering within micromaps (Fig. 2c,d; Fig. 4b-d).
2. Presentation of additional fields of view supporting the columnar organization of micromaps (Extended Data Fig. 3).
3. Distributions shown of responses to all color conditions for every cell class, and analyses of changes observed with structured vs. unstructured stimuli (Extended Data Fig. 6).
4. Analyses of overall color tuning in response to structured vs unstructured stimuli, accounting for inter- and intra-axis contributions (Extended Data Fig. 7; vector-based classification of cells in Extended Data Table 2).
5. Analyses of color tuning metrics in blobs vs. interblobs, and in cells that responded only to structured or unstructured stimuli (Extended Data Fig. 8).
6. Analyses of color microdomain segregation/overlap observed with structured vs unstructured stimuli (Extended Data Fig. 9e).
7. Expanded discussion of color tuning in response to structured vs unstructured stimuli in main text (pp. 5-6).
8. Additions and clarifications to Methods.

Reviewer #1 (Remarks to the Author):

In this paper, Chatterjee and colleagues use two-photon calcium imaging to investigate the micro-organization of color tuning in superficial layers of monkey V1. This is a unique study that will contribute important insights to our understanding of the functional organization of V1. The paper will likely be of broad interest for the general neuroscience community. I have a number of minor suggestions, mostly regarding the data analysis.

Minor suggestions:

1.1. *For most of the paper, two-photon data are analyzed as dF/F , with the exception of the pixel-based maps. If there is a reason for this choice, it would be useful to discuss it; otherwise the authors may wish to plot dF/F instead of dF .*

We show ΔF versions of the pixel-based maps since they provide the clearest view of responsive cell clusters. These figures are used only to introduce the reader to an early, minimally derived presentation of color micromaps and as a point of reference for the more

derived cell-based maps, not for any quantification. We now discuss this choice of ΔF maps in the Methods [pp. 17-18]:

We used ΔF instead of $\Delta F/F$ responses for clarity of presentation, since the denominator in $\Delta F/F$ maps can introduce noise from areas of low resting fluorescence, like vascular shadows. Also, ΔF maps are more appropriate for comparing signals between cell bodies and neuropil, making it easier to visualize the puncta of cell bodies in higher magnification images (e.g., Fig. 1c).

1.2. *In addition, the normalization of responses across cone contrast appears problematic, as OGB responses are known to follow a saturating nonlinearity. The effects of this nonlinearity on the conclusions should be discussed.*

The reviewer is correct to raise the issue of OGB nonlinearity, but it likely had a minimal effect on our conclusions since $\Delta F/F$ responses were generally quite small in this study. We added a discussion of this to the Methods [pg. 18]:

OGB-1 AM responses are known to follow a saturating nonlinearity^{42,43}, which can complicate the interpretation of contrast-normalized responses. However, sublinear saturation effects become prominent only when OGB-1 AM signals are greater than approximately half the maximum possible signal, which can be 300% $\Delta F/F$ or more in many preparations⁴². The responses in this study were much smaller (rarely above 20% $\Delta F/F$), placing these responses well below the onset of significant saturating nonlinearities.

1.3. *It would strengthen the results if the authors could show maps for more of the imaging regions included in the data analyses.*

We agree. In the new Extended Data Fig. 3, we show 10 imaging planes from two animals with complete micromaps (seven new planes, and three previously used in the main figures). The figure also supports the observation of similar clusters across depths, as requested by the reviewer (comment 1.9, below). Maps from the other two animals were obtained from only 1-2 depths per animal and were already shown in Extended Data Fig. 5b,c, so the addition of these new maps should give the reader a comprehensive view of the dataset.

1.4. *Currently, the cell-based analysis may overemphasize distinctions between cells, as each cell is assigned to one color category only based on its preferred response. It might be better to quantify color preference with a vector in color space that represents not only the best response, but overall color tuning. Similarly, it would be useful to include an analysis of potential changes in color preferences across stimulus types.*

This suggestion led to an extensive series of new analyses addressing the broad question of “potential changes in color preferences across stimulus types” (see Extended Data Figs. 6-8).

We kept the main text’s preferred-response metric for classifying cells using uniform stimuli for two reasons. First, this method captures what the reader can see by eye in the various ΔF maps shown in the manuscript (Fig 1b,c; Fig 2a), particularly the segregation and clustering of chromatic domains. Second, discrete classes simplified certain analyses, like subclustering of color domains (Fig. 2f). However, now we justify this choice and show that a new, vector-based classification gives nearly identical populations (Extended Data Fig. 7d,h). We added a discussion of this in the Methods [pp. 18-19]:

Discrete classes based on best response to uniform conditions simplified the interpretation of cell-based maps, both visually and in analyses like subclustering within micromaps (Fig. 2f). They were justified by comparing with an alternative, vector-based quantification of color tuning, in which we plotted responses on a plane defined by orthogonal red/green and blue/yellow axes (red/green coordinate from equation 2, blue/yellow from equation 3, in Extended Data Fig. 7). Each cell’s overall color tuning in this plane was given by the angle θ between the positive x-axis and the vector from the origin to the cell’s location in the plot. For uniform stimuli, the two classification methods largely resulted in the same populations (Extended Data Fig. 7d, h; see also Extended Data Table 2). However, the preferred-condition method was not appropriate for classifying cells based on responses to bars, independent of uniform stimuli, since intra- and inter-axis mixing with bars precluded unambiguous assignment to a single color class by single best response (see Extended Data Figs. 6 and 7). To independently classify cells based on structured stimuli, we used the vector analysis method described above (Extended Data Table 2).

More importantly, this new measure of overall color tuning allowed us to quantify changes across stimulus types, leading to a richer and, frankly, surprising analysis of color responses. As we write in the expanded Discussion [pp. 5-6]:

Color tuning was also highly sensitive to stimulus spatial structure. Spatially uniform conditions evoked simple responses from cells in chromatic micromaps (e.g., Fig. 2a; time courses in Extended Data Fig. 2), similar to those in the dorsal lateral geniculate nucleus, as can be seen across the population in the distributions of responses to each color condition (Extended Data Fig. 6a) as well as in plots of overall tuning incorporating contributions from both red/green and blue/yellow color axes (Extended Data Fig. 7c,d and g,h; see Methods). There was some *inter*-axis mixing consistent with previous work¹⁸, particularly blue/yellow contributions to red/green cell classes. However, responses to uniform stimuli were relatively pure in the sense of minimal *intra*-axis contributions, defined as responses to conditions from the opposite or sign-reversed end of a class’s dominant axis, like green (+M, M–L, –L) responses from cells in the red (–M, L–M, +L) class (Extended Data Fig. 7).

This purity of color tuning to uniform stimuli was sharpened by surprising class-specific suppression to individual color conditions (Extended Data Fig. 6a), distinct from the general suppression to all conditions shown in Fig. 3. The distributions of responses to intra-axis (green) conditions were significantly suppressed in red cells, as were most red responses in green cells. The distributions of responses to all six red and green conditions were significantly suppressed in the blue (+S, –L–M) class. And at the extreme of inter- and intra-axis suppression, seen in the yellow (–S) cell class, the median response to every condition except –S was below baseline. All classes showed suppressed responses to the luminance condition (L+M), except for the 16 cells (1.3% of 1,242 coactive cells) making up the luminance class.

With drifting bars, class-specific suppression disappeared. Median responses significantly increased to almost every condition in every cell class (Extended Data Fig. 6b), resulting in significantly more inter- and intra-axis mixing (Extended Data Figs. 7 and 8a). Responses to red/green conditions increased far more than to blue/yellow across all classes (Extended Data Fig. 6d), eroding the clear distinctions between classes seen with uniform stimuli (compare Extended Data Fig. 7 d,h and l,p). Using an alternative, vector-based classification of color tuning (since the best-response classification used with uniform stimuli does not capture intra- and inter-axis mixing; see Methods and Extended Data Table 2, right columns), we found that this unbalanced gain between the two color axes caused a redistribution of cells classified as blue/yellow to red/green, particularly away from the yellow (–S) class, which saw the smallest increase in response to structured stimuli.

1.5. *While the increase in the number of responsive neurons for bars versus uniform stimuli is obvious, there currently is no statistical test for it.*

We added a statistical test for this. [pg. 4]: “Neurons responding to bars were significantly clustered (Fig. 4b; $P = 4.9 \times 10^{-4}$, Wilcoxon signed-rank test), and there were more responsive cells on average to bars than to uniform conditions (Fig. 4c; paired $t_{1,241} = 3.03$, $P = 0.011$).”

1.6. *The text states that color modules overlap more strongly for bars versus uniform stimuli, but does not provide quantification for this statement.*

We quantify this now in Extended Data Fig. 9e.

1.7. *The characteristic length measure appears problematic when multiple clusters are present in the same imaging region. Neuron pairs spanning clusters will contribute long inter-pair distances, which will form a second peak in the distance histogram. The relative height of the peaks will depend on the size of the clusters. As the characteristic length is determined at half peak height, it can be affected by the presence and relative size of the other peaks. In addition, it captures the differences in map structures induced by bars versus uniform stimuli rather*

indirectly: The increase in responsive neurons with bar stimuli will likely result in filling in the 'middle distance' range of the distance histogram, rather than changing the initial part of it. The only reason for an increased characteristic length is then that this filling in is strong enough to move the location of the first peak, rather than demonstrating the overall larger region. It might be good to add a second metric that more directly captures the compactness of the responsive regions.

We agree that secondary peaks can shift the distance to half-maximum of the first peak. While this actually made our original measure *more* conservative (i.e, secondary peaks should shift the characteristic length to the right, bringing it closer to the characteristic length of the shuffled control), the reviewer raises valid concerns, so we re-analyzed the full dataset using a more standard, published measure of clustering (virtually the same as the analysis in reference 26).

Our original measure of clustering took the pairwise distribution of distances between responsive cells and compared the characteristic length of the first peak (distance to half-max) with the characteristic length of a shuffled control. However, we did not normalize the measured distribution by the number of *all possible* cells pairs. The number of possible cell pairs increases in proportion to distance, then falls off as the distance approaches the size of the field of view. Thus, the more appropriate analysis is to plot the fraction of all possible cell pairs that are responsive cell pairs, as a function of pairwise distance. The shuffled distribution (also normalized by all possible pairs) now becomes flat, with no peaks. If cells are significantly clustered, the normalized distribution should start well above the shuffled line (more clustered than by chance), then dip below as the distance increases, and finally approach the line again. This is exactly what we find in our data (Fig. 2c,d; compare with ref. 26, Fig. 2b). We define the characteristic length in this analysis to be the point at which the measured distribution crosses the shuffled line, which is a more straightforward metric. Significance was tested by calculating a clustering index, as in references 24-26. We kept the additional, neighborhood tests of clustering unchanged (Fig. 2e,f), since the concerns outlined above do not apply.

The reviewer notes that it may not be appropriate to compare characteristic lengths for maps obtained with uniform vs bar stimuli. We completely agree and further argue that this “filling in the middle distance range of the histogram” also makes clusters that were separate with uniform stimuli merge into single clusters as more cells become responsive with bars. We now do the more appropriate analysis of demonstrating significant compactness or clustering with bars (Fig. 4b), and then showing that these clusters contain significantly more cells than those obtained with uniform stimuli (Fig. 4c).

1.8. *Suppression of responses seems to occur for all colors based on the example cells. Is that generally the case?*

Only by definition, which we have clarified in the Methods [pg. 17]: a “further selection [criterion] of . . . maximum $\Delta F/F < 0$ in response to all 10 color conditions for suppressed cells.” However, to the reviewer’s point, the new analyses of color tuning revealed significant class-

specific suppression as well, which we show in Extended Data Fig. 6 with asterisks marking significantly suppressed response distributions.

1.9. *To be able to correctly interpret the result shown in Figure 1G, either maps for each imaging depth should be shown, or the number of cells included per depth need to be stated (otherwise it cannot be ruled out that the result is simply due to the majority of cells coming from one imaging depth only).*

We think the reviewer meant Figure 2g, and we agree. We now show separate maps for each plane used in that figure (Extended Data Fig. 3a-e), as well as those from another animal in which multiple depths were imaged (Extended Data Fig. 3g-l).

1.10. *Clarification for methods: How large was the stimulus in visual degrees, and how large were the estimated receptive fields of the recorded neurons? At which eccentricities were the experiments done approximately?*

We have added this information to the Methods [pg. 16]: “Stimuli covered approximately 23 x 30 visual degrees. We did not measure receptive field sizes of recorded neurons, but recording locations corresponded to ~6-8 degrees eccentricity and receptive field centers ~2 degrees in diameter^{38,39}.”

1.11. *Clarification for methods: Was the sequence of stimulus conditions randomized? If not, are there concerns regarding adaptation?*

The sequence was not randomized. We are not aware of a study that quantifies chromatic adaptation in macaque V1 as a function of presentation time and cone contrast, but most cortical adaptation studies use many 10s of seconds of a continuous adapting stimulus (e.g., high contrast achromatic gratings) before interleaving test stimuli.

We have clarified that the stimuli were nonrandomized in the Methods [pg. 16]: “Each condition was followed by a blank of the same duration (3 s for uniform, 3-6 s for bars and gratings), and all stimuli were presented sequentially and repeated 10-20 times. Conditions were not randomized, but the long duration of neutral gray between each condition likely mitigates adaptation concerns, particularly with respect to general micromap structure and how the spatial content of stimuli affects functional organization.”

1.12. *Abstract: The abstract claims a columnar organization of color tuning. Two-photon imaging is limited to upper layer 2/3 in macaques, and can therefore not demonstrate a full columnar organization. It would be better to either clearly state the limit in the abstract, or to rephrase columnar to clustered/patchy organization.*

We now state that the data are limited to upper layer 2/3: “Imaging from multiple depths in upper layer 2/3 suggests that this ‘micromap’ organization is columnar.”

Reviewer #2 (Remarks to the Author):

This paper describes a 2-photon calcium imaging study that addresses a long-standing issue in color neurophysiology: Are the spatially lowpass, chromatically opponent neurons of the macaque primary visual cortex enriched in the cytochrome oxidase-enriched blobs? The answer appears to be an unequivocal "yes". I enjoyed reading and learning from this paper.

Major comments:

2.1. *Which neurons contribute to color vision is an unresolved issue. For this reason, I find the authors' use of the word "chromatic" (e.g. in the context of "chromatic cells" and "chromatic maps") confusing. My reading of the paper suggests that a luminance-tuned cell that doesn't contribute to color vision but responds well to uniform stimuli would have been included in the "fine-scale map of chromatic preference". Is this so? Were such low-pass, non-opponent cells observed? An analysis of cone weights would be useful. Do non-opponent neurons cluster spatially in the cortex similarly to opponent neurons?*

There was a surprisingly small number of such low-pass, luminance-preferring cells in our population: 16 cells out of 1,242 coactive cells (from the 12 fields of view used in comparing responses to bars and uniform stimuli; see Extended Data Fig. 6a, bottom panel), and 64 cells out of our total population of 3,365 responsive to uniform stimuli (see Extended Data Table 1). With so few luminance-preferring cells per field of view, we can say very little about their clustering.

The analysis of cone contributions suggested by the reviewer served as impetus to fully analyze overall color tuning in response to color conditions presented as spatially uniform vs. drifting bar stimuli (Extended Data Figs. 6-8). Since this was also suggested by the first reviewer, please see our response to comment 1.4 above for a discussion of these results.

2.2. *Among cone-opponent neurons, is there a tendency for the sign of the S-cone weight to match the sign of the M-cone weight, as electrophysiological experiments have found? Does the degree of cone-opponency differ between neurons in the blobs and the interblobs when probed with bars or gratings?*

We believe the reviewer is referring to the results in Conway, 2001 (ref 18). Interestingly, we found that this result holds for uniform stimuli but not for bars [pg. 32]: “There was a weak negative correlation between red/green and blue/yellow inputs obtained with spatially uniform

stimuli ($r = -0.08$, $n = 1226$, $P = 0.007$; data from scatterplots **c** and **g**), supporting previous work¹⁸ that suggested M- and S-cone responses were often aligned in sign. This correlation disappeared with bars ($r = 8 \times 10^{-4}$, $n = 1226$, $P = 0.98$; data from scatterplots **k** and **o**)."

We analyzed color tuning metrics for cells in blobs and interblobs in response to both uniform and bar stimuli, and as we note in the Discussion [pp. 6-7]: "Cells responding to both uniform and bar stimuli clustered near blobs, but bars-only cells were distributed evenly between blobs and interblobs. We observed small or no significant differences in color tuning metrics between coactive cells in blobs and interblobs, or between coactive cells and either bars-only or uniform-only cells (Extended Data Fig. 8). These were largely overlapping distributions." The single most important factor determining the degree of a cell's cone-opponency was the spatial content of the stimuli (Extended Data Fig. 8a).

2.3. *How much of the spatial structure in the photomicrographs is due to neural activation and how much is due to loading of the dye? The Methods mention that the dye spread over ~1 mm, and I assume that the dye penetration was roughly uniform over this area, but an analysis of the spread of the dye relative to the size of the activated cell clusters, if possible, would be welcome.*

Unfortunately, OGB does not survive fixation and cytochrome oxidase histology, so we are unable to definitively address the issue of dye spread vs. cell clusters. The ~1 mm figure in the Methods comes from *in vivo* scanning as we searched for candidate fields of view with the brightest fluorescence and least overlying vasculature before starting the stimulus protocol. However, the spread of labeled cells always extended beyond the chosen field of view, and dye penetration was roughly uniform in the injection area (we are not aware of any OGB study in which patchy labeling is discussed as an issue). Note that the apparent patchiness of label in the example field of view in Fig. 1a is almost entirely due to vascular shadows and edge vignetting related to the optics of our scope.

One line of evidence already in the manuscript strongly suggests that observed cell clustering was not an artifact of uneven spread of dye. As shown by the distribution of characteristic lengths in Fig. 2d (left), clusters in response to spatially uniform stimuli were invariably much smaller than the 765 x 765 μm field of view. With drifting bars, these same clusters expanded and more cells were significantly responsive, implying that the tightly circumscribed clusters observed with uniform stimuli were physiological in origin (e.g., Fig. 4 and Extended Data Fig. 9).

Minor comments:

2.4. *Throughout the paper, stimuli are referred to as "cone-isolating" but four of them—L-M, L+M, -L-M, and M-L—do not isolate a single cone type.*

We agree and have changed all occurrences of "cone-isolating", except for those referring to actual cone-isolating gratings, to either "cone-specific" or "color" conditions.

2.5. *The final sentence of the abstract and of the discussion refer to "dynamic switching". Temporal changes in V1 activity are not addressed in this study, so it seems strange to conclude the paper with a speculation about them. I realize that different spatio-chromatic patterns of light entering the eye produce different patterns of cortical activity, and that such patterns change over time, but I do not see how the results of this study add to this picture. Along the same lines, I am unsure what the word "robust" in the last sentence of the abstract is intended to mean.*

We agree that the use of "dynamic" was ill-advised, since it does suggest a temporal component to our study. We kept the word "switch" in its common, non-dynamic usage (switch between x and y), which the maps do in response to structured vs. unstructured stimuli, switching between different populations of active cells. Also, we agree that the word "robust" is content-free. The last line of the abstract now reads: "We conclude that V1 has a[n] ~~robust and dynamic~~ architecture for color representation that switches between blobs and a combined blob/interblob system based on the spatial content of the visual scene." We also removed "dynamic" from the last line of the discussion.

2.6. *"The statistics of natural scenes also support a preferentially low-spatial-frequency color system." I do not see how the statistics of natural images motivate a color system that is tuned for lower spatial frequencies than the luminance system.*

We have removed the line and reference.

2.7. *Referring to the L, M, and S cones as "red", "green" and "blue", even parenthetically, is counterproductive.*

We agree that using L, M, and S exclusively does not assume too much sophistication on the part of the reader. The first occurrence of cone types now reads [pp. 2-3]: "The responsive cells in this field of view carry hallmarks of low-level chromatic processing, including largely opponent signals from long (L) and medium (M) wavelength-sensitive cones when both are large enough to detect, along with strong, often dominant signals from short (S) wavelength-sensitive cones."

2.8. *Cell-based maps: Please state the number of cell classes (I believe the answer is 4) and clarify how the categorization was done. I think most of the relevant information is in the legend to Extended Data Figure 2, but this material is sufficiently important that it should be moved to the Methods or to one of the figure legends in the main text.*

Agreed. The following text has been added to the Methods [pg. 18]:

Cells were assigned to color classes based on their preferred spatially uniform condition, using positive $\Delta F/F$ responses normalized by each condition's cone contrast (see Extended Data Fig. 2). Suppressed responses were not normalized for any analyses. There were five classes: red cells responded best to +L, L-M, or -M; green to +M, M-L, or -L; blue to +S or -L-M; yellow to -S; and luminance to L+M. These designations also applied to the stimulus conditions themselves (e.g., +L, L-M, -M are referred to collectively as red conditions). Class names reflect the approximate color of their conditions. For analyses comparing responses to spatially uniform and drifting bar stimuli (Extended Data Figs. 6-8), initial class designations based on uniform stimuli were carried through to bar analyses, keeping comparisons within the same cell populations.

2.9. *"0.06%; non-unique ..." I don't understand this.*

Yes, this was confusing, but it no longer applies. The larger problem was that a proper comparison of general suppression across stimulus types should have been done only in the same population of cells, which was not the case before.

We redid the analysis of suppression to uniform and bar stimuli using the same 12 fields of view that we used for all other comparisons between the two stimuli (e.g., Fig. 4, Extended Data Figs. 6-8), and the sentence now reads [pg. 5]: "Notably, general suppression disappeared completely with structured stimuli (1,660 suppressed cells to uniform stimuli; 9 to bars)."

2.10. *Why are t-tests used for some comparisons and Wilcoxon tests for others?*

We used t-tests only for paired comparisons of means. For almost everything else, we used non-parametric comparisons of medians (e.g., Wilcoxon) simply because many of the distributions were skewed enough that parametric tests of means would not have been appropriate (e.g., the CO distributions in Fig. 3c). However, our p-values were generally quite small due to the large size of our population, and we confirmed (not shown) that the result of every hypothesis test in the manuscript remained unchanged regardless of our choice of parametric vs. non-parametric tests.

2.11. *How was the image in Figure 1A obtained? Does it reveal where the dye-loaded cells were located, independently of their stimulus preferences? Does it show the locations of all responsive cells? Why there are fewer visible cells in Figure 1b?*

The image in Fig. 1a was obtained by averaging a large number of acquisition frames, now clarified in the Methods [pg. 17]: "Reference two-photon images were generated by averaging > 100 frames of motion-corrected data for each plane (e.g., Fig. 1a) . . ." It was included to give the reader a qualitative sense of dye loading and the size of our fields of view. There is a very small component of stimulus-evoked activity in Fig. 1a, mostly from the cluster of responsive

cells seen in the boxed area of Fig. 1b (the contribution of neuronal activity to the brightness of any pixel is likely much less than 10%, the approximate $\Delta F/F$ response from the boxed cluster of cells). For the most part, Fig. 1a is dominated by the baseline fluorescence of the dye and should be used as a proxy for dye-loaded cells independent of responsive cells.

There are fewer visible cells in Fig. 1b because it is a ΔF map. Since it shows only *change* in fluorescence in response to stimulus conditions, only active cells are revealed [pg. 17]: “Each pixel-based map is a ΔF image, defined as blank image subtracted from condition image, where blank image is the average frame immediately preceding all stimulus conditions. Condition images were averaged over stimulus repetitions and maps were spatially smoothed (gaussian, $\sigma = 2.8 \mu\text{m}$).”

2.12. *Figure 2. I am confused about the meaning of the gray curves. The integral of the gray curves and the black histogram are similar, suggesting that similar numbers of cells were used to create each of them. Yet, the text says that the curves were obtained by "repeatedly shuffling the positions of all identified cells within the field of view". Also, a single gray curve follows the black histogram closely. This presumably is not a result of random permutation, but this should be made clear.*

We agree that those gray curves were ambiguous. However, the figure as been replaced with a new analysis of clustering (Fig. 2c,d; please see our response to comment 1.7, above, for a discussion of the new figure).

2.13. *Was the 2 Hz presentation of the spatially uniform and bar stimuli synchronized to the 2 Hz data acquisition?*

No, and we now state this in the Methods [pg. 16]: “Stimulus presentation was not synchronized with data acquisition.” Since calcium responses obtained with OGB are sustained due to slow calcium kinetics (e.g., traces in Extended Data Fig. 2), stimulus timing should not significantly affect the quantification of responses.

2.14. *Extended Data Figure 5 legend: "significantly responding to > 8 conditions". "Stimulus condition" could mean spatial structure, but I do not think that it does here. Does this refer to 8 of the 10 basic stimulus conditions (+L, +M, -L, -M, L-M, M-L, -L-M, L+M, -S, +S)?*

Correct, and we have clarified the text to reflect this [pp. 36-37]: “Some cells outside of color domains (bottom row) responded to most or all of the 10 color conditions, regardless of cone type or sign. There were 181 such cells (significantly responding to > 8 color conditions; Tukey HSD, $P < 0.05$) out of a population of 4,205 responsive cells from 12 FOVs (4.3%).”

Reviewers' Comments:

Reviewer #1:

Remarks to the Author:

The revised manuscript addresses all of my concerns, and presents important and novel findings regarding the representation of color in monkey primary visual cortex. I would strongly recommend it for publication. My only suggestion is to make many of the results for the structured stimuli main figures (instead of extended figures) to more accurately reflect the fundamental nature of these findings (this pertains in particular to extended data figures 6 and 7).

Reviewer #2:

Remarks to the Author:

A new contribution of the revised manuscript is that observation that color tuning is strongly dependent on the spatial structure of the stimulus. These results are broadly consistent with those of Thorell et al. 1984 and Lennie et al. 1990, but are inconsistent with those of Johnson et al. 2001. Perhaps a sentence on this discrepancy is warranted.

"Spatially uniform conditions evoked simple response...". "simple responses" could be interpreted as meaning "phase sensitive" (as opposed to complex responses) which is not what I think is intended here.

Figure 4b. "dotted lines ± 1 SD across permutations". I am not sure what this refers to.

The vector-based quantification of color tuning could be described more clearly. I presume that the preferred color direction was the direction of the vector sum of responses to individual stimuli, but the text is unclear on this point. I did not find the explanation of the preferred color direction in terms of the cell's location useful.

Extended Data Figure 9. If the 2-D correlation coefficient is the same correlation coefficient, then dropping the "2-D" would make this easier to understand. If it's not, it should be explained.

I was unsure how to interpret Extended Figure 9F and did not find a reference to it in the main text.

We thank the reviewers again for supporting the work and their helpful suggestions. Our responses, and all changes to the main text, are indicated in red.

Reviewer #1 (Remarks to the Author):

The revised manuscript addresses all of my concerns, and presents important and novel findings regarding the representation of color in monkey primary visual cortex. I would strongly recommend it for publication. My only suggestion is to make many of the results for the structured stimuli main figures (instead of extended figures) to more accurately reflect the fundamental nature of these findings (this pertains in particular to extended data figures 6 and 7).

We agree that the new structured stimulus results are important, and we seriously considered the reviewer's reasonable suggestion to shift those Supplementary Figures to the main text. We decided against this because the manuscript is ultimately about color *maps*, and how these maps contain micromaps of cone specificity, align with cortical modules, and shift with the spatial content of the stimulus. Note that every main figure is built on map exemplars (intentionally). We felt a greater emphasis on new *physiology* in the main figures may obscure this straightforward narrative somewhat.

Also, we appreciate having the space in the Supplementary Information to adequately explain our analyses, as attested by the length of our Supplementary Figure legends. Moving the figures to the main text, and distributing the text of the old legends throughout the main manuscript and Methods, could be counterproductive. Better to keep it all together.

Reviewer #2 (Remarks to the Author):

2.1 *A new contribution of the revised manuscript is that observation that color tuning is strongly dependent on the spatial structure of the stimulus. These results are broadly consistent with those of Thorell et al. 1984 and Lennie et al. 1990, but are inconsistent with those of Johnson et al. 2001. Perhaps a sentence on this discrepancy is warranted.*

We agree and included a sentence using language very similar to the reviewer's [pg. 6]:

"In general, the observation that color tuning strongly depends on the spatial structure of the stimulus is broadly consistent with earlier electrophysiological studies that examined this question^{7,22}, with exceptions³¹."

2.2 *"Spatially uniform conditions evoked simple response...". "simple responses" could be interpreted as meaning "phase sensitive" (as opposed to complex responses) which is not what I think is intended here.*

We agree and cut the word "simple", since the rest of the sentence does not require it:

"Spatially uniform conditions evoked ~~simple~~ responses from cells in chromatic micromaps . . . similar to those in the dorsal lateral geniculate nucleus."

2.3 Figure 4b. "dotted lines ± 1 SD across permutations". I am not sure what this refers to.

The dotted lines show the SD of mean clustering indices of the position-shuffled control FOVs (for visually comparing shuffled values to the real clustering indices shown as circles).

The reviewer is correct that this was poorly worded on our part. We made two edits: 1) we expanded the Methods section dealing with this analysis; and 2) we show the SD of clustering indices for *all* position-shuffled controls (not just for the mean indices of each FOV) and updated the legends and Source Data for Figs. 4b and 2d accordingly. This makes more sense as a view of the spread of shuffled values. Note that this is only for display purposes and affects no conclusions, since the SD is not used in our statistical tests of clustering.

Methods [pg 13]:

"An index of clustering²⁴⁻²⁶ (Figs. 2d and 4b) was determined for each FOV by first calculating the median absolute pairwise distance between responsive cells across all pairs located within 200 μm (approximately the mean characteristic length over all FOVs), and then doing the same calculation for each of 1,000 position-shuffled controls. The clustering index for each FOV was then defined as the median of the median pairwise distances of all position-shuffled controls (the grand median) divided by the median pairwise distance of the observed data. If this ratio is 1, the observed clustering is taken to be similar to a random distribution. The more the ratio exceeds 1, the greater the clustering."

Figure 4b legend:

". . . gray line, mean clustering index for position-shuffled controls; dotted gray lines, ± 1 SD ($n = 12,000$)."

Figure 2d legend:

". . . gray line, mean clustering index for position-shuffled controls; dotted gray lines, ± 1 SD ($n = 17,000$)."

2.4 The vector-based quantification of color tuning could be described more clearly. I presume that the preferred color direction was the direction of the vector sum of responses to individual stimuli, but the text is unclear on this point. I did not find the explanation of the preferred color direction in terms of the cell's location useful.

The reviewer's presumption is correct. We have added the vector sum explanation, keeping the original as well since we want to refer the scatterplot [Supp pg. 13]:

"The preferred color direction θ of each cell is the direction of the vector sum of red/green (*interRG*) and blue/yellow (*interBY*) responses (i.e., the direction of the vector from the origin to the cell's location on the scatterplot)."

2.5 Extended Data Figure 9. If the 2-D correlation coefficient is the same correlation coefficient, then dropping the "2-D" would make this easier to understand. If it's not, it should be explained.

This was a good catch. We had explained the calculation in the Methods section: “Two-dimensional correlation coefficients (Supplementary Figure 9e) were computed with the ‘corr2’ algorithm in Matlab,” but we had not included a reference to Methods in the text. We have corrected this [Supp pg. 18]:

“We quantified the increased overlap of microdomains observed with drifting bar stimuli (as seen in **b**) by calculating the two-dimensional correlation coefficient for each pair of KDE images from a given FOV (same FOVs as in **c**), for a given stimulus type (uniform, bars; see Methods).”

Calculating correlation coefficients between 2-D arrays (like image pairs) does use a different algorithm than the 1-D case, and are usually named as such in most statistical packages like R and Matlab. We have maintained that convention.

2.6 *I was unsure how to interpret Extended Figure 9F and did not find a reference to it in the main text.*

Another good catch. We had referenced Supplementary Figure 9f in the 9b legend, but not in the main text. We have corrected this:

“Even so, the chromatic maps defined with uniform stimuli were maintained with bars (Supplementary Figure 9; single-condition maps in Supplementary Figure 9a,b from same field of view as Fig. 4a), but each individual subdomain underwent an expansion as well, creating more overlap of previously distinct subdomains and new ‘hotspots’ (Supplementary Figure 9e; example single-cell time courses in Supplementary Figure 9f).”

As indicated, Supplementary Figure 9f shows example time courses of responses to bars, from cells in different subdomains of micromaps. We think it is important to show at least a few actual response traces, since the rest of the bar analysis is far more derived.